# Intima-Media Thickness and Pulsatility Index of Common Carotid Arteries in Acute Ischaemic Stroke Patients with Diabetes Mellitus

**DOI:** 10.3390/jcm12010246

**Published:** 2022-12-29

**Authors:** Olivier Bill, Michael V. Mazya, Patrik Michel, Tiago Prazeres Moreira, Dimitris Lambrou, Ivo A. Meyer, Lorenz Hirt

**Affiliations:** 1Department of Neurology, Lausanne University Hospital and University of Lausanne, 1011 Lausanne, Switzerland; 2Department of Neurology, Karolinska University Hospital, 171 77 Stockholm, Sweden; 3Department of Clinical Neuroscience, Karolinska Institutet, 141 86 Stockholm, Sweden; 4Department of Old Age Psychiatry and Psychotherapy, University of Bern, 3012 Bern, Switzerland

**Keywords:** stroke, diabetes mellitus, duplex sonography, pulsatility index, intima-media thickness, acute ischemic stroke

## Abstract

Ultrasonographic parameters such as the common carotid artery (CCA) pulsatility index (PI) and CCA intima-media thickness (IMT) have been associated with an increased mortality and risk of recurrent stroke, respectively. We hypothesized that these ultrasonographic parameters may be useful for monitoring diabetic patients after an acute stroke. We analysed retrospective data of consecutive acute ischaemic stroke patients from the ASTRAL registry who underwent pre-cerebral ultrasonographic evaluation within 7 days of symptom onset. We compared clinical, demographic, radiological and ultrasonographic parameters in diabetic versus non-diabetic patients (univariable and multivariable analyses) and the association of these parameters with CCA PI and CCA IMT. We analysed 1507 carotid duplex ultrasound examinations from patients with a median age of 74 years. Cardiovascular co-morbidities, including hypertension, hypercholesterolemia, obstructive sleep apnoea syndrome, higher body-mass index (BMI) and peripheral artery disease, were associated with diabetes mellitus (DM). Diabetics were more often under antiplatelet therapy and had atrial fibrillation at admission. Diabetic patients showed an increased CCA PI and IMT in line with more atherosclerotic changes on acute CTA compared to non-diabetic patients. Taking IMT as the dependent variable in a second analysis, DM, higher age, hypertension, smoking and CCA PI were associated with higher IMT. Taking CCA PI as the dependent variable in a third analysis, DM, higher age and higher NIHSS at admission were associated with higher CCA PI values. Increased IMT was also associated with higher PI. We show that CCA PI and IMT are higher in diabetic patients in the first week after an initial stroke.

## 1. Introduction

Diabetes mellitus (DM) is a major risk factor for cerebrovascular disease [1,2]. Patients with DM have increased stroke mortality [3] and are known to develop more clinically evident [4] and progressive [5] atherosclerosis. Ultrasonography is commonly used for assessing atherosclerotic burden, carotid or vertebral artery stenosis, plaque anatomy and carotid artery intima-media thickness (IMT), and for detecting patent foramen ovale [6] in stroke patients [7]. Increased IMT has been shown to be a risk factor for stroke [8,9]. Stroke risk estimation is significantly improved by an assessment of plaque characteristics [10] using ultrasonography methods [11,12]. DM patients have higher IMT values [13,14], a larger plaque burden [15] and plaques that are more prone to rupture [16].

The Pulsatility Index (PI) [17] in the common carotid artery (CCA) reflects distal resistance to cerebrovascular blood flow [18,19,20] and has been used to assess the haemodynamic impact of intracranial and extracranial stenosis [21,22]. This cerebrovascular resistance is associated with cardiovascular morbidity and mortality in stroke patients [23,24]. Vascular disease indicators on transcranial Doppler (TCD) progress faster in DM patients [25,26,27], whose plaque morphology also displays a more severe profile [14,28]. This suggests that ultrasonographic parameters could be used to predict symptomatic cerebrovascular disease (CVD) in diabetics. Diabetes is associated with higher internal carotid artery and middle cerebral artery PI [29]. We hypothesise that, at the time of acute ischaemic stroke (AIS), hospitalised diabetic patients have different ultrasonographic parameters, such as IMT and CCA PI values, compared to non-diabetic patients. We aimed to define the clinical and neurosonological profile of diabetic stroke patients, as well as to assess the clinical and radiological factors associated with CCA IMT and PI.

## 2. Materials and Methods

The present study is a retrospective, single-centre analysis of ultrasound examinations in consecutive AIS patients referred to the cerebrovascular ultrasound laboratory during their hospital stay at the Centre Hospitalier Universitaire Vaudois (CHUV) in Lausanne from 14 October 2004 to 31 December 2014. Patient data were obtained from the ASTRAL database [30]. Duplex sonography data were extracted from the Lausanne Doppler Registry (LaDoRe) [22]. The Duplex sonography examination considered for this study was the closest in time after the AIS but was always within 7 days of the index event. Transient ischaemic attacks designated by either the time-based or tissue-based definition (depending on the availability of magnetic resonance imaging) were excluded. Patients with AIS and a prior or newly discovered diagnosis of DM were compared with non-diabetic AIS patients. Duplex sonography examinations were performed on Acuson Sequoia 512 (probes: 8L5 for cervical, Siemens Medical Solutions, Mountain View, CA, USA), ATL HDi 5000 Advanced Ultrasound Instrument (probes: L12-5 for cervical, Philips Medical Systems, Eindhoven, the Netherlands) or IU 22 (probes L9-3 for cervical, Philips) devices. IMT was measured on longitudinal 2D-echography images of the carotid bifurcation, 1 cm proximally from the bifurcation [8], either manually or using the Q-lab software (Philips). We used IMT of the CCA, as it predicts future ischaemic stroke recurrence better than the IMT of the internal carotid artery [31]. CCA PI was calculated as originally described [17], i.e., PI = (peak systolic velocity–end diastolic velocity)/(mean velocity). We collected baseline clinical parameters, such as sex, age, type of DM, cerebrovascular disease risk factors and stroke characteristics, such as clinical localisation, recurrence, aetiology and severity. We obtained ultrasonographic measurements, including the number of intracranial and extracranial stenoses within and outside the stroke territory, CCA peak systolic velocity, CCA end-diastolic velocity, CCA PI, CCA IMT and plaque characteristics. We classified plaques as homogenous, heterogeneous, regular, irregular, eroded, ulcerated, hyper-, hypo-, iso- or an-echogenic. Velocity criteria were used to define stenosis ranging from 50% to 70%, with an assessment of the presence of collateral flow through the ophthalmic artery and anterior cerebral artery for stenosis above 70%, as recommended by Von Reutern et al. [32]. Patients were excluded from the analysis according to ASTRAL criteria [30] and for the following reasons: incomplete ultrasonography examination, no interpretable exam, missing critical data in the patient health record (stroke side, DM status, unknown territory), age < 18 years, isolated glucose intolerance without a diagnosis of DM. Vertebrobasilar strokes were also excluded, as well as patients with a history of previous strokes, bilateral moderate to severe carotid stenosis or occlusion, carotid stenting or endarterectomy, because of heterogeneity of the vessel wall and hemodynamic effects.

### Statistical Analysis

The study consisted of two parts: first, we performed descriptive univariable analysis (univariable analysis 1) for baseline clinical and ultrasonographic data, as well as plaque morphology, comparing patients with and without DM. Second, we performed multivariable logistic regression analyses (multivariable analyses 1, 2a and 2b) to assess independent variables associated with DM, CCA IMT and CCA PI as dependent variables, respectively. For continuous variables, we calculated the median and interquartile range. For categorical variables, we calculated proportions by dividing the number of events by the total number of patients, excluding missing or unknown cases. Before conducting the multivariable analyses, we performed an imputation of missing data using the chained equations method. In this process, five imputed datasets were generated. Each dataset was analysed separately, and the significant associations of the response with the covariates were determined using stepwise methods. In all multivariable analyses, we made allowance for the possible stochastic association of CCA PI measurements on the same patient, using a compound symmetric error structure. All analyses were performed using the R statistical package (version 3.5.3).

## 3. Results

A total of 13,719 Duplex sonography examinations were available in the registry. We analysed 1507 carotid ultrasonographic exams performed on 1424 patients with acute ischaemic stroke of the anterior territory. Of these, 320 (22.5%) had DM, of which 270 (18.9%) had a known diagnosis and 51 (0.4%) were newly discovered. Seventy-five patients (0.5%) had two strokes and thus two examinations, and eight patients had three examinations. All had a Duplex sonography examination within seven days after onset, and the median time for examination after admission was one day. Three hundred and thirty-nine patients (23.8%) had a previous clinical cerebrovascular event, 83 (0.6%) had a bilateral carotid stenosis, and 21 (0.2%) patients were excluded because of high-grade (≥70%) bilateral stenosis. The median BMI was 25, and 209 patients (14.7%) were obese. In the univariable analysis (Table 1 and Appendix A), diabetic stroke patients were older, more often male, more often already taking anti-platelet drugs on admission, had a higher BMI, had more cardiovascular co-morbidities such as hypertension, hypercholesterolemia, obstructive sleep apnoea syndrome, peripheral artery disease, and also had coronary heart disease and atrial fibrillation more often. In addition, diabetic patients had a higher atherosclerotic burden on admission imaging with CT angiography. On Duplex sonography examination, the IMT values and PI indices were higher in the CCA and ICA in DM patients, and plaques were more often hyperechogenic. The baseline stroke severity, age and TOAST categories did not show statistical differences.

In the multivariable analyses (Table 2), a higher BMI and obstructive sleep apnoea syndrome were independently associated with DM. Antiplatelet therapy at admission, anti-hypertensive drugs and lipid-lowering medication were associated with DM stroke patients, as was peripheral artery disease and atrial fibrillation. Regarding radiological investigations, DM was associated with atherosclerotic changes on admission angio-CT and a higher CCA IMT and CCA PI in the Duplex sonography.

In the multivariable analysis 2a (see also Table 2), with IMT as the dependent variable, a higher age, DM, hypertension, smoking and higher CCA PI were associated with a higher IMT. On the contrary, female sex, cardio-embolic and undetermined stroke mechanisms and right-side examination of the CCA were associated with a lower IMT.

In the multivariable analysis 2b, with PI as the dependent variable, patients with DM, an older age, higher admission NIHSS, anti-hypertensive drugs, higher IMT, significant extracranial vessel stenosis and right-side examination of Duplex sonography parameters were associated with higher CCA PI values. Female sex, smoking, atrial fibrillation, a higher BMI and significant atherosclerosis were negatively associated.

As Appendix A, we provide predicted average CCA PI values for patients with a specific characteristic (e.g., diabetic or not, smokers or non-smokers, etc.), keeping the other variables at their mean value (Appendix A).

## 4. Discussion

Our study shows that diabetic stroke patients have a higher burden of cerebrovascular risk factors and associated findings on Duplex sonography examination, i.e., a higher Intima-Media Thickness (IMT) and Pulsatility Index (PI) of the common carotid artery. These anatomic and rheologic Duplex sonography parameters were independently associated with DM, as well as age and hypertension (higher IMT and PI) and female sex and cardioembolic strokes (lower IMT and PI).

It was previously shown in several stroke cohorts that diabetic stroke patients have more cerebrovascular risk factors [22]. In our study, more patients were already under lipid-lowering medication at the time of the stroke, which reflects the fact that DM appears earlier in life than the initial stroke in the elderly population in our registry (median age was 73 years). In particular, the diabetic stroke population also shows signs of early vascular aging (a higher pulsatility and thicker vessel walls reflected by higher PI values and IMT measurements in the large pre-cerebral arteries, despite already being under cerebrovascular prevention treatments). An association between DM and IMT was previously described in a smaller cohort [33]. In our work, diabetic stroke patients have a higher CCA PI on the initial Duplex sonography, a finding that is in line with previous observations in the carotid artery in a non-stroke population [29] and was also described in the kidney [34] and retina [33,35] of diabetic patients. PI of the CCA provides a more robust index for examining patients, as it does not rely as much on the examiner; it is independent of the angle of insonation [36].

Higher IMT has previously been reported to be associated with DM and age [37,38]. IMT was lower in female stroke patients in our study. This has been described earlier, but not specifically in acute stroke [39]. This might indicate that unisex cut-off values for this widely used marker should be taken with caution in female patients.

High CCA PI is also associated with age, hypertension, high-grade stenosis and NIHSS severity. Age and hypertension have been linked to higher PI values [40,41] and other markers of vascular resistance in stroke populations [42]. High-grade internal carotid artery stenosis increases resistance in upstream arteries, and an association with increased CCA PI is logical. Higher NIHSS scores have been linked to elevated PI in the MCA [43] but have not been described as linked to PI in the CCA. Smoking showed higher IMT but lower PI, and on the contrary, a right-side exam showed lower IMT and higher PI. The hypothetical link with smoking is that smoking is a progressive burden on the vessels, diminishing the elasticity of the vessel wall [44] without changing much of the flow in pathophysiological backgrounds [45]. Blood flow and IMT differences between the right and left carotid have been described [46]. The right carotid tends to have less output and thus carries less burden on the vessel wall and the flow decreases with age [47].

The strengths of this work are the detailed consecutive registration of the patient data in ASTRAL combined with a detailed Duplex sonography assessment in routine practice, providing more than 70 variables per patient. The large number of patients with complete datasets makes the results robust and generalizable and allows us to identify 12 independent factors associated with CCA PI. Our data was collected over a long period (more than ten years), with pre-specified and standardized data collection (scales, definitions and neurovascular imaging methods). Consistent with previous ASTRAL publications, the patient sample includes both a primary and tertiary referral population to our stroke centre. The limitations are those of a retrospective, observational and single-centre study. In addition, we did not record blood cholesterol levels or detailed data on the duration, type or control status of diabetes before the stroke, which could have allowed for the identification of further associations or better explanations of our findings. Further, the results were not adjusted for treatment of DM or secondary prevention drugs used between admission and ultrasound examination, nor were they adjusted for the stroke side. Multiple testing was not accounted for. As our results show that most associations are significant at the 1% level, corrections for multiple testing would not influence the study conclusions. ASTRAL only contains patients admitted within 24 h of stroke onset or with a last proof of good health and may therefore not be representative of patients with subacute strokes. Further work could help to monitor the atherosclerotic burden along the diabetic patient’s lifelong disease progression and help identify collateral flow patterns in diabetic patients. Lower IMT was found in females, which raises the possibility of a lower cut-off for this subgroup.

## 5. Conclusions

Diabetic stroke patients show differences in arterial resistance and flow properties and show a higher burden of atherosclerosis on their precerebral vessel walls at the first stroke compared with non-diabetic patients.

In our study, CCA PI and IMT were useful markers of these properties in a routine acute stroke evaluation. We show that CCA PI and IMT are higher in diabetic patients in the first week after an initial stroke.

## Figures and Tables

**Table 1 jcm-12-00246-t001:** Univariable analysis (only selected variables shown, see also Appendix A).

	Diabetic (n = 320)	Non-Diabetic (n = 1187)	*p* Value
Baseline characteristics			
Age	75 (16)	73 (21)	0.06
NIHSS	7 (11)	7 (11)	0.97
BMI	27.6 (6.5)	25 (6)	<0.001
Sex (female)	122 (38.1%)	525 (44.3%)	0.048
Acetylsalicylic acid	137 (43.1%)	371 (31.6%)	<0.001
Oral antihypertensive	240 (75.7%)	668 (57.0%)	<0.001
Oral lipid-lowering agent	143 (44.8%)	300 (25.5%)	<0.001
Insulin	65 (20.5%)	0 (0.0%)	<0.001
Risk factors			
Hypertension	273 (85.3%)	831 (70.2%)	<0.001
Hyperlipidaemia	271 (85.2%)	891 (75.6%)	<0.001
Smoking	78 (24.8%)	278 (24.1%)	0.80
Atrial fibrillation	110 (34.4%)	316 (26.8%)	0.008
Peripheral artery disease	41 (13.0%)	67 (5.7%)	<0.001
Obstructive apnoea syndrome	28 (9.3%)	38 (3.4%)	<0.001
TOAST group			
Large artery atherosclerosis (LAS)	56 (18.6%)	184 (16.1%)	0.58
Radiology			
CTA/MRA atherosclerosis	193 (83.5%)	663 (69.4%)	<0.001
CTA/MRA significant pathology in ischaemic territory	150 (63.6%)	608 (63.1%)	0.90
Recurrence			
None	246 (88.5%)	947 (89.2%)	0.28
Ultrasonographic parameters			
Right CCA PI	1.64 (0.4) n = 314	1.56 (0.4) n = 1160	<0.001
Left CAA PI	1.59 (0.4) n = 313	1.52 (0.5) n = 1159	<0.001
Right CCA IMT	0.90 (0.4) n = 302	0.80 (0.3) n = 1135	<0.001
Left CCA IMT	1.00 (0.4) n = 298	0.90 (0.4) n = 1129	<0.001

Abbreviations. BMI: body mass index, CCA: common carotid artery, CTA: computed tomography angiography, IMT: intima-media thickness, MRA: magnetic resonance angiography, NIHSS: National Institutes of Health Stroke Scale, PI: pulsatility index, TOAST: Trial of Org 10172 in Acute Stroke Treatment. Data are presented as n (%) for categorical data or median (interquartile range) for continuous data.

**Table 2 jcm-12-00246-t002:** Multivariable analyses with DM (analysis 1), common carotid artery pulsatility index (analysis 2a) and intima-media thickness (analysis 2b) as the dependant variables, respectively.

DM as Dependent Variable (Analysis 1)	OR	95% LL	95% UL	*p* Value
BMI	1.12	1.09	1.15	<0.001
Sleep apnoea	1.67	1.09	2.56	0.019
Antiplatelets	1.32	1.06	1.63	0.011
Under hypertensive drugs at admission	1.37	1.09	1.73	0.008
Under lipid-lowering drugs at admission	1.52	1.23	1.87	<0.001
Peripheral artery disease	1.76	1.28	2.43	0.001
Atrial fibrillation	1.29	1.05	1.58	0.016
Atherosclerotic changes on acute CTA	1.53	1.12	2.09	0.010
IMT	1.48	1.08	2.02	0.014
CCA PI	1.54	1.20	1.99	0.010
**CCA IMT as Dependent Variable (Analysis 2a)**	**Coef.**			
Age	0.0058	0.0048	0.0068	<0.001
DM	0.0395	0.0072	0.0718	0.016
Hypertension	0.0344	0.0016	0.0673	0.040
Smoking	0.0543	0.0217	0.0869	0.001
CCA PI	0.0323	0.0022	0.0625	0.035
Sex (female vs. male)	−0.0452	−0.0724	−0.0181	
TOAST mechanism (cardio. vs. LAS)	−0.0566	−0.0970	−0.0163	0.006
TOAST mechanism (undet. vs. LAS)	−0.0465	−0.0861	−0.0070	0.021
Side (right vs. left)	−0.0514	−0.0675	−0.0353	<0.001
**CCA PI as Dependent Variable (Analysis 2b)**	**Coef.**			
DM	0.0664	0.0286	0.1042	0.01
Age	0.0050	0.0035	0.0064	<0.001
NIHSS at admission	0.0042	0.0020	0.0064	<0.001
Antihypertensive drugs at admission	0.0358	0.0015	0.0700	0.041
IMT	0.0576	0.0086	0.1066	0.021
Significant stenosis on extracranial vessels on admission CTA	0.0496	0.0104	0.0888	0.014
Side (right vs. left)	0.0420	0.0207	0.0634	<0.001
Sex (females vs. males)	−0.0706	−0.1015	−0.0396	<0.001
Smoking	−0.0421	−0.0789	−0.0054	0.025
Atrial fibrillation	−0.0397	−0.0751	−0.0043	0.028
BMI	−0.0044	−0.0084	−0.0005	0.029
Acute CTA atherosclerotic changes	−0.0524	−0.1014	−0.0033	0.037

Abbreviations. BMI: body mass index, cardio: cardio-embolic, CCA: common carotid artery, Coef: coefficient; CTA: computed tomography angiography, IMT: intima-media thickness, NIHSS: National Institutes of Health Stroke Scale, PI: pulsatility index, TOAST: Trial of Org 10172 in Acute Stroke Treatment, undet: undertimined.

## Data Availability

The data that support the findings of this study are available from the corresponding authors on reasonable request.

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
