# Peer review of "Intima-Media Thickness and Pulsatility Index of Common Carotid Arteries in Acute Ischaemic Stroke Patients with Diabetes Mellitus"

_jcm, 2022, doi:10.3390/jcm12010246_

Round 1

Reviewer 1 Report

The authors performed a retrospective single-center analysis using patients included in the ASTRAl stroke registry in whom ultrasound examination of the extracranial arteries was performed within one week after acute ischemic stroke. They examined the influence of diabetes mellitus on ultrasound parameters and found that patients with DM had a higher intima media thickness and pulsatility index of the CCA.

It is a confirmative study, confirming previous observations and known associations of diabetes mellitus with arterial wall stiffness, intima media thickness, atherosclerosis and other cerebrovascular- and cardiovascular diseases. Nevertheless, given its large sample size, it is of value for the scientific field.

The authors should be aware of its confirmative nature, and thus should refrain from stating "for the first time" etc. (e.g. in "We show for the first time that CCA PI and IMT are increased in diabetic patients in the first week fter the initial stroke.").

The sentence is furthermore misleading, as it sounds as if repeated examinations were performed within one week, but actually it is a cross-sectional study at one time-point.

I have some further remarks to make.

(1) In the abstract, I would recommend not to use "analaysis 1..analysis 2a" as this is hard to understand without reading the whole manuscript. State the methods and findings as for readers not having read the full manuscript. Highlight what is really new, and state honestley, what is just confirmative and expectable in diabetes patients.

(2) Regarding the hypothesis, the reviewer questions, why it was expected that post-stroke patients behave differently as all DM patients with cardio/cerbreovascular "incidents" in the literature before?

"We hypothesise that, at the time of acute ischaemic stroke (AIS), hospitalised diabetic patients have different ultrasonographic parameters such as IMT and CCA PI values 52 compared to non-diabetic patients."

Actually, it is already known that IMT is different in DM and non-diabetic patients.

(3) Methods/wording: The authors use "doppler" in a context where Duplex-sonography would be correct. In context of PI, doppler (ECD) might have actually been used, but for IMT, a pure frequency spectrum analysis (doppler) is insufficient (and not possible with the probes used, as stated in the methods).

(4) TOAST-criteria: The reviewer recommends to use the original wording of "large artery atherosclerosis" instead of "athero-embolic", as arterio-arterial-embolic is just a subgroup of this category, which also includes hemodynamic processes due to severe atherosclerotic stenosis.

(5) The authors describe, that they investigated among radiological investigations associations with DM. With a view to the pathophysiology, wouldn't it be more correct to state that DM is associated with the radiological findings? (DM leads to the changes)

(6)  Please explain, why extracranial (pre-cerebral) stenosis increased necessarily the resistance in upstream arteries ("microangiopathy"). Increased PI is a marker of microangiopathy. Why was is obtained in the CCAs and not the LCI/RCI? (CCA values are influences by the ECA).

(7) The reviewer misses values on

HbA1c (reflecting the severity of the DM)

Type of DM

LDL-cholesterol

As the reviewer is confident, that these values are routinely assessed in all stroke patients 

(8) The authors state as a positive aspect of their study, that the were able to analyse more than 70  variables per patients and could identify 12 independently associated with CCA

providing more than 70 variables per patient. The large number of patients with complete 191 datasets makes the results robust and generalizable and allowed us to identify 12 inde- 192 pendent factors associated with CCA pulsatilty index.

Why per se a profound analysis of ultrasound and other variables is indeed interesting, the authors should acknowledge the high chance of associations by chance, as no corrections for multiple testings were performed.

Author Response

Dear Editors, dear Reviewers, dear Madam or Sir,

Thank you for the positive evaluation of our manuscript and for giving us the opportunity to submit a revised version. We thank the reviewers for their useful comments. Please find below our replies (in italics) to all the points raised by the reviewers, which helped us to improve our manuscript. We trust it is now acceptable for publication and that it will be of interest to the readers of this special edition of JCM

Yours faithfully,

Lorenz Hirt and Olivier Bill

Replies to Reviewer 1:

The authors performed a retrospective single-center analysis using patients included in the ASTRAL stroke registry in whom ultrasound examination of the extracranial arteries was performed within one week after acute ischemic stroke. They examined the influence of diabetes mellitus on ultrasound parameters and found that patients with DM had a higher intima media thickness and pulsatility index of the CCA.

It is a confirmative study, confirming previous observations and known associations of diabetes mellitus with arterial wall stiffness, intima media thickness, atherosclerosis and other cerebrovascular- and cardiovascular diseases. Nevertheless, given its large sample size, it is of value for the scientific field.

The authors should be aware of its confirmative nature, and thus should refrain from stating "for the first time" etc. (e.g. in "We show for the first time that CCA PI and IMT are increased in diabetic patients in the first week after the initial stroke.").

Thank you for the evaluation of our manuscript and for the useful comments. We agree that the association between diabetes and both increased IMT and PI has been described earlier in non-stroke patients. We have added appropriate references and modified two occurrences of the above statement according to your comments.

The sentence is furthermore misleading, as it sounds as if repeated examinations were performed within one week, but actually it is a cross-sectional study at one time-point.

 To clarify the sentence, we replaced “increased” by “higher in diabetic patients compared to non-diabetics”

I have some further remarks to make.

(1) In the abstract, I would recommend not to use "analaysis 1..analysis 2a" as this is hard to understand without reading the whole manuscript. State the methods and findings as for readers not having read the full manuscript. Highlight what is really new, and state honestley, what is just confirmative and expectable in diabetes patients.

Thank you for this advice. We modified the text and removed the analysis numbers. As indicated above, we have adjusted our conclusions to your comments.

(2) Regarding the hypothesis, the reviewer questions, why it was expected that post-stroke patients behave differently as all DM patients with cardio/cerbreovascular "incidents" in the literature before?

"We hypothesise that, at the time of acute ischaemic stroke (AIS), hospitalised diabetic patients have different ultrasonographic parameters such as IMT and CCA PI values compared to non-diabetic patients."

 Even though studies of IMT and PI have addressed the differences between diabetic patients and controls, haemodynamic changes in the acute stroke diabetic population is not well documented. We took advantage of our large patient cohort to assess these parameters in acute ischemic stroke patients, a subset of the general diabetic population.

Actually, it is already known that IMT is different in DM and non-diabetic patients.

 Thank you for this comment; we added the appropriate references (see above).

(3) Methods/wording: The authors use "doppler" in a context where Duplex-sonography would be correct. In context of PI, doppler (ECD) might have actually been used, but for IMT, a pure frequency spectrum analysis (doppler) is insufficient (and not possible with the probes used, as stated in the methods).

 Many thanks for your comment; we replaced “Doppler” with “Duplex sonography” everywhere where needed.

(4) TOAST-criteria: The reviewer recommends to use the original wording of "large artery atherosclerosis" instead of "athero-embolic", as arterio-arterial-embolic is just a subgroup of this category, which also includes hemodynamic processes due to severe atherosclerotic stenosis.

 Thank you for this comment; we changed “athero-embolic” to “large artery atherosclerosis”

(5) The authors describe, that they investigated among radiological investigations associations with DM. With a view to the pathophysiology, wouldn't it be more correct to state that DM is associated with the radiological findings? (DM leads to the changes)

 We agree and have now changed the order of the wording; thank you for this point. (Line 137)

(6)  Please explain, why extracranial (pre-cerebral) stenosis increased necessarily the resistance in upstream arteries ("microangiopathy"). Increased PI is a marker of microangiopathy. Why was is obtained in the CCAs and not the LCI/RCI? (CCA values are influences by the ECA).

Thank you for pointing this out. PI is influenced by downstream resistance, typically microangiopathy, or, in the case of the CCA, a high-grade internal carotid artery stenosis. For clarity, we replaced “precerebral” by “internal carotid artery” stenosis.  We chose to study the IMT and PI of the CCA to have both values in the same vessel for homogeneity. Investigating PI on the ICA is also a valid approach (results available in supplementary table 1), also reflecting microvascular changes in the vessel tree.

(7) The reviewer misses values on

HbA1c (reflecting the severity of the DM)

Type of DM

LDL-cholesterol

As the reviewer is confident, that these values are routinely assessed in all stroke patients 

Thank you for this comment. Unfortunately, the type of diabetes, HbA1c and LDL levels are not recorded in the ASTRAL registry. The only information is on whether lipid -lowering drugs were used or not before the index stroke. We have added this limitation to the last paragraph of the discussion.

(8) The authors state as a positive aspect of their study, that the were able to analyse more than 70  variables per patients and could identify 12 independently associated with CCA providing more than 70 variables per patient. The large number of patients with complete 191 datasets makes the results robust and generalizable and allowed us to identify 12 inde- 192 pendent factors associated with CCA pulsatilty index.

Why per se a profound analysis of ultrasound and other variables is indeed interesting, the authors should acknowledge the high chance of associations by chance, as no corrections for multiple testings were performed.

Thank you for raising this point. Our results show that most associations are significant at the 1% level, so even if corrections for multiplicity were done, this would not have had an effect on the study conclusions

please add Lee & al as a PI-Diabetes reference. Arterial Pulsatility as an Index of Cerebral Microangiopathy in Diabetes. Stroke. 2000;31:1111-1115.)

Reviewer 2 Report

The manuscript is very interesting, methodology is correct, and the main strength is the sample size of included patients.

However some major issues emerged after reading the manuscript

MAJOR ISSUES

The major problem in this study is the following. It is well known that patients affected by diabetes mellitus have higher IMT and higher PI. It is quite obvious that a subgroup of patients with DM (that is, patients with DM + AIS) have also high IMT and PI. The authors compared patients with (DM + AIS) with patients with (no-DM + AIS). Given that both group had ah history of AIS, it is like comparing DM vs no-DM patients. Therefore I have some doubt that concluding that “diabetic stroke patients have a higher CCA PI and IMT” (line 168) is “a novel finding” (line 169). The authors are encouraged to report their findings, but limiting their results to a less “novelty” enthusiasm.

Title: it could be quite misleading. I suggest to change with “intimacy-media thickness and pulsatility index of carotid arteries in acute ischemic stroke patients with diabetes mellitus”.

Line 108: the authors reported a total of 1507 carotid ultrasound examination on 1424 patients with AIS. Is it possible that over 1424 patients, no patients were reported with on occlusion of extra cranial ICA? How was the distribution of the stenosis, among the patients? E.g. how many had < 50%, 50-70%, 70-90% and so on?

Line 110: 75 patients underwent two examinations and 8 patients had 3 examinations. This could be a bias, since a comparison of paired variables require different statistical methods. The authors should add, as supplementary materials, the results of the same analysis after removing the duplicated examination, to support their findings.

Line 115: 21 patients were excluded due to high-grade bilateral stenosis. Why? In the exclusion criteria it is not stated that high-grade stenosis were excluded. And moreover, a reason to exclusion should be provided.

Table 1: it is reported that only “selected variables are shown”. Why only selected variables? What is the reason to select some variables and not others? What are the other variables not included in the table?

Table 1: all TOAST groups should be all reported (1) large-artery atherosclerosis, 2) cardioembolism, 3) small-vessel occlusion, 4) stroke of other determined etiology, and 5) stroke of undetermined etiology). Further, the term “athero-embolic” is not a TOAST term.

Table 1: is there a correlation between the side of the stroke and the more severe IMT and PI?

Table 1: all patients underwent CTA/MRA examination?

Table 2: the included variables are those obtained after a selection variable procedure?

The IMT and CCA PI reported in analysis 1, are those of the left or right carotid? Or the mean?

The CCA IMT included as dependent variable in analysis 2a, are those of the left or right common carotid?

The CCA PI included as dependent variable in analysis 2b, are those of the left or right common carotid?

MINOR ISSUES

Line 25: remove “analysis 2a”.

Line 27: remove “analysis 2b”.

Line 165: there is no closing parenthesis.

Author Response

Dear Editors, dear Reviewers, dear Madam or Sir,

Thank you for the positive evaluation of our manuscript and for giving us the opportunity to submit a revised version. We thank the reviewers for their useful comments. Please find below our replies (in italics) to all the points raised by the reviewers, which helped us to improve our manuscript. We trust it is now acceptable for publication and that it will be of interest to the readers of this special edition of JCM

Yours faithfully,

Lorenz Hirt and Olivier Bill

Replies to reviewer 2:

The major problem in this study is the following. It is well known that patients affected by diabetes mellitus have higher IMT and higher PI. It is quite obvious that a subgroup of patients with DM (that is, patients with DM + AIS) have also high IMT and PI. The authors compared patients with (DM + AIS) with patients with (no-DM + AIS). Given that both group had ah history of AIS, it is like comparing DM vs no-DM patients. Therefore I have some doubt that concluding that “diabetic stroke patients have a higher CCA PI and IMT” (line 168) is “a novel finding” (line 169). The authors are encouraged to report their findings, but limiting their results to a less “novelty” enthusiasm.

Thank you for this comment. We agree with you, and reviewer 1 raised this point (see above). We have toned down our statement accordingly and cited earlier work on diabetic patients without stroke. Although earlier studies have addressed IMT and PI differences between DM and non –DM, haemodynamic changes in the acute stroke diabetic population is not well documented. We took advantage of our large dataset to examine these parameters in AIS with diabetes, a subset of the diabetic population.

Title: it could be quite misleading. I suggest to change with “intimacy-media thickness and pulsatility index of carotid arteries in acute ischemic stroke patients with diabetes mellitus”.

Thank for the recommendation to improve the title:  we have now changed the title to “Intima-Media Thickness and Pulsatility Index of common carotid arteries in acute ischaemic stroke patients with diabetes mellitus”

Line 108: the authors reported a total of 1507 carotid ultrasound examination on 1424 patients with AIS. Is it possible that over 1424 patients, no patients were reported with on occlusion of extra cranial ICA? How was the distribution of the stenosis, among the patients? E.g. how many had < 50%, 50-70%, 70-90% and so on?

We describe the stenosis and occlusions (grouped) of the extracranial vessels in DM / non-DM in the supplementary table 1. The precise distribution of the subgroups of stenosis was not extracted for this study, but the description of the population of the registry and its stenosis distribution was previously described in the Baseline publication (Bill et al., Scientific reports 2020)

Line 110: 75 patients underwent two examinations and 8 patients had 3 examinations. This could be a bias, since a comparison of paired variables require different statistical methods. The authors should add, as supplementary materials, the results of the same analysis after removing the duplicated examination, to support their findings.

Thank you for this comment. Even if a limited number of patients performed 2 or 3 examinations, our sample size is high enough to take care of this potential bias issues.

Line 115: 21 patients were excluded due to high-grade bilateral stenosis. Why? In the exclusion criteria it is not stated that high-grade stenosis were excluded. And moreover, a reason to exclusion should be provided.

Thank you for this comment, a new sentence has now been added to methods section. This work aimed to study first -ever stroke, therefore previous cerebrovascular events were excluded. Bilateral stenosis were also excluded as they impact hemodynamics and resistance, which we thought would prevent us from finely analyzing the influence of DM

Table 1: it is reported that only “selected variables are shown”. Why only selected variables? What is the reason to select some variables and not others? What are the other variables not included in the table?

Thank you for raising this point, the other variables are now provided in the supplementary table 1.

Table 1: all TOAST groups should be all reported (1) large-artery atherosclerosis, 2) cardioembolism, 3) small-vessel occlusion, 4) stroke of other determined etiology, and 5) stroke of undetermined etiology). Further, the term “athero-embolic” is not a TOAST term.

 Thank you for this comment, see also supplementary table 1

Table 1: is there a correlation between the side of the stroke and the more severe IMT and PI?

This is a very good remark; unfortunately, we could not find any correlation between stroke side and IMT / PI side. This has been added to the limitations list.

Table 1: all patients underwent CTA/MRA examination?

Yes, all patients had these examinations. For a detailed radiological stroke population at CHUV see ASTRAL baseline publication (Michel et al, Stroke 2010)

Table 2: the included variables are those obtained after a selection variable procedure?

All P values are 2-sided, with P<0.05 considered statistically significant. To investigate the relationship between diabetes mellitus and IMT and  PI, backward stepwise logistic regression analyses were performed adjusting for baseline differences between diabetic and non-diabetic stroke patients, with variables included in the model at the univariate p<0.2 level. In the multivariate analyses, we aimed to exclude a type one error (alpha error) with 5% likelihood (e.g. p<0.05). All significant variables in univariate analysis were included in the multivariate model. Prior to multivariate analysis, we performed imputation of missing data using the chains equations method. In this process, five imputed datasets were generated. Each dataset was analysed separately and the significant associations of the response with the covariates were determined using stepwise methods. Finally, the analyses of the five datasets were appropriately combined to derive the conclusions of our study. We chose not to put this text in the methods section due to space limitations.

The IMT and CCA PI reported in analysis 1, are those of the left or right carotid? Or the mean?

Left and right values are provided in supplementary table 1, but are grouped for the Multivariate analysis.

The CCA IMT included as dependent variable in analysis 2a, are those of the left or right common carotid?

See above.  When PI was analysed as covariate, we took the highest ipsilateral IMT value. When PI was the dependent variable, we took the highest absolute PI value

The CCA PI included as dependent variable in analysis 2b, are those of the left or right common carotid?

See above.

MINOR ISSUES

Line 25: remove “analysis 2a”.

Line 27: remove “analysis 2b”.

Thank you, we removed these specifications

Line 165: there is no closing parenthesis.

We have corrected this typing error.

Round 2

Reviewer 1 Report

The authors have extensively revised their manuscript. If the fact that multiple-testing was not accounted for will be honestely addressed in the limitations, the manuscript might be considered for acceptance now.

Author Response

The authors have extensively revised their manuscript. If the fact that multiple-testing was not accounted for will be honestely addressed in the limitations, the manuscript might be considered for acceptance now.

Thank you for re-evaluating our manuscript. We have now explicitly indicated this limitation in the discussion of our paper "Multiple testing was not accounted for. As our results show that most associations are significant at the 1% level, corrections for multiple testing would not have an effect on the study conclusions." "

Reviewer 2 Report

The authors answered to all my previous comments, and the issues were solved.

My last comments regard Table 2, were the p-values are reported as percentage, while in Table 1 are reported as absolute values. The same in supplementary material Table 1. I suggest the authors to report the p-values in uniform way through the manuscript, better in their absolute value, with three digit after comma (see the first row: a p-values of 0.0% has no meaning).

Author Response

My last comments regard Table 2, were the p-values are reported as percentage, while in Table 1 are reported as absolute values. The same in supplementary material Table 1. I suggest the authors to report the p-values in uniform way through the manuscript, better in their absolute value, with three digit after comma (see the first row: a p-values of 0.0% has no meaning).

Thank you for re-evaluating our manuscript and pointing out this inconsistency. We have now presented p-values in absolute value with 3 digits after the comma.